# Psychosocial Workers and Indigenous Religious Leaders: An Integrated Vision for Collaboration in Humanitarian Crisis Response

**David William Alexander [1,\*] and Tatiana Letovaltseva [2,3]**

1   Centre for Trauma, Asylum and Refugees, University of Essex, Colchester CO4 3SQ, UK
2   Research Unit of Pastoral and Empirical Theology, KU Leuven, 3000 Leuven, Belgium; tatiana.letovaltseva@mil.be
3   The Royal Military Academy, 1000 Bruxelles, Belgium
\*   Correspondence: dalexaa@essex.ac.uk

**Abstract:** Indigenous religious leaders can be the most trusted organic helping agents within vulnerable communities, but often lack orientation to the language and paradigms of the mental health and psychosocial support (MHPSS) professionals responding to their communities after a crisis. Similarly, MHPSS professionals work within paradigms which do not always match the indigenous world views of the vulnerable people they seek to help and therefore can undermine community stability while attempting to provide a service. In parallel, the spiritual care offered by indigenous religious leaders does not always optimally intersect with evidence-based MPHSS interventions, although it is highly likely that both approaches to care provide important benefits to the community, some of which are missing or underemphasized in one or the other. Training approaches designed to orient religious leaders to the work of MHPSS are usually funded and delivered by MHPSS professionals and tend to leverage MHPSS assumptions and portray MHPSS interventions as the most important lines of effort in care. This may leave religious leaders feeling uncertain of their ability to contribute to multi-disciplinary efforts without migrating away from their own foundational assumptions about humanity, illness, and wellness. Often missing from the field is a parallel effort in training which offers MHPSS professionals insight into the efficacy of indigenous spiritual interventions of various kinds and how working alongside indigenous religious leaders can aid them in protecting against the well-known pathologizing tendencies present in their own models of care. The authors are experienced in working during and after community crisis with both MHPSS professionals and indigenous religious leaders and offer an integrated vision for combined training and combined support planning that may facilitate collaboration after crisis in vulnerable communities.

**Keywords:** epistemological vigilance; mental health and psychosocial support; religious actors; spiritual care; trauma; teleology of suffering

## 1. Introduction

After political violence swept through a rural village in her region, a nurse and psychosocial worker named Bojana was deployed by her government to care for vulnerable survivors in a hastily constructed camp for internally displaced persons.[1] Bojana is a specialist well-trained to identify persons most likely to need formal psychiatric support in the aftermath of acute adversity but was unprepared in this case for the depth and scope of the religious language used by survivors to convey their experiences and their outlook. She noted that a religious leader was among the group of survivors and that many of the other survivors turned to the religious leader for support and guidance. Bojana did not share the religious framework of these survivors. For this reason, she felt ill-equipped to attempt a translation of their religious language into terminology which related sufficiently to her psychiatric assessments. Bojana approached the religious leader for advice on the

matter, but he seemed to her to express a mild distrust of her expertise and a disinterest in working together within her approach. Instead, he suggested that she follow her own practices, while he separately practiced his.

Andreanos is a traditional religious leader[2] who lives and works in an underdeveloped coastal community which has survived several natural disasters in recent years.[3] Many families have been locally displaced because of these disasters, and helpers from at least three different organizations have offered psychosocial support resources as a part of larger responses. Although he is not trained in a mental health discipline, Andreanos has decades of experience providing care and support to families in his community. Observing the psychosocial workers and their approach among his people, he noticed many important contributions that they were making. However, he also noticed what he felt were short-comings, in that they emphasized certain aspects of well-being over others in a manner not traditionally shared by his people. He briefly aspired to collaborate with these psychosocial workers but felt that they expected him to adopt their own approach and orientation to do so, which left him conflicted.

The experiences of Bojana and Andreanos represent different facets of a common dilemma in mental health and psychosocial support (MHPSS) initiatives among religious survivors of natural disaster and political violence. Indigenous religious leaders are some-times the most trusted organic helping agents within vulnerable communities, but they often lack an orientation to the language and paradigms of the mental health workers who respond after a crisis. Even when they do have such an orientation, they may resist or contest certain aspects of these paradigms. This can obviously frustrate the potential for a collaboration which may broadly benefit the community.

From another perspective, MHPSS professionals work within paradigms which do not always match the traditional world views of the vulnerable people they seek to help. This is further complicated by evidence in the psychosocial literature that even the best MHPSS response frameworks risk oversimplifying and pathologizing the experiences of survivors (Papadopoulos 2021; Cummings 2012; Pupavac 2002). In some cases, MHPSS professionals feel they might benefit from the perspectives and input of indigenous religious leaders, i.e., to better contextualize their efforts, but simultaneously worry these religious leaders may hinder their efforts in some way (Osafo 2016; Slife and Reber 2009). However, they may also be unfamiliar with the unique nature of the contributions of religious leaders to total well-being—at least as seen from the community's perspective—and therefore miss the opportunity to consider whether such contributions could provide a vital supplement to their own approaches (Alexander 2023; Seddon et al. 2011).

Bojana and Andreanos met each other face to face in the summer of 2016 and both recognized an opportunity immediately.[4] Bojana finally met an indigenous religious leader in the region of her deployment who was happy to collaborate with her and offer her better access to community understanding and trust—if she was able to make room for him to work from his own understandings of illness, wellness, and flourishing. In Bojana, Andreanos met a psychosocial worker who was seeking to work with him to better understand his people and their needs, if he was willing to allow her some leeway to work within the frameworks that she believed would help him and his people, although they originated outside of the community. Together they formed a partnership which lasted and extended far beyond their brief time working together in a single location. Indeed, they met monthly by video teleconference for nearly three years between 2016 and 2018, enriching each other's perspective and providing mutual support and peer supervision to each other from a distance.

We the authors have been working with religious actors and psychosocial workers in various regions and contexts for a combined 25 years. We have examined many efforts which claim to bring both together into a mutually respectful, professionally egalitarian, and therapeutically coherent approach to support during community crisis. However, few have been able to accomplish what Bojana and Andreanos accomplished together. In our view, there is a fundamental reason for their success. Bojana and Andreanos did not begin

with an exploration of how they could practically work together, but with an exploration and appreciation of their core differences.

## 2. Making a Beginning

The various tensions in knowing and perceiving between psychological and religious helping discourses are well-known and widely addressed in the psychosocial literature (cf. Hodge et al. 2020; Abu-Raiya et al. 2016; Paloutzian and Park 2013). Strange, then, that few efforts to bring religious actors and psychosocial workers together therapeutically begin with a careful examination of these tensions and the underlying epistemological gaps these tensions represent. Epistemology may be defined as the inquiry into the nature, conditions, and extent of human knowledge (Papadopoulos 2006).

Bojana was willing to accept Andreanos' initial terms for working together, i.e., that he be allowed to work from his own epistemological foundation, rather than adopting the assumptions implicit in modern psychological theories and approaches to care. Specifically, he felt that there would be meaningful differences between the two, including answers to such fundamental questions as: What is a person? How and why do people change? What is the destiny of humans? What leads to true human flourishing? Andreanos wished to stay close to an approach to care for human persons which maintained answers to such questions that flowed from the religious beliefs of his community. She suggested that they initially spend time discussing the differences in their orientations—beginning with what they felt were the most important beliefs, priorities, and practices they leveraged as helpers—and each try to find admirable aspects of the other's approach.[5]

There are many practical reasons for avoiding this foundational process of seeking to understand epistemological differences between MHPSS professionals and religious leaders who wish to collaborate. It may seem counter-productive to explore differences when the primary aim is working together. Exploring foundations may seem like a poor use of time, especially when time is scarce, and needs in the field are so large as to be overwhelming. Not least of the common reasons for avoiding such exploration is the fear that the differences which are discovered will prove to be—for one or both parties—insurmountable, and collaboration no longer attractive. However, no matter what the reason, avoidance of this process of exploration is likely to be an error with persistent consequences, because such differences will not be rendered insignificant simply because they are out of sight. There is a long arc of evidence within the literature, likely beginning with Karpman (1968) and more recently expanded by Papadopoulos (2013, 2016, 2018, 2021), that after-action reviews of therapeutic effectiveness are most successful when they review not only action but the therapeutic positioning of the actors, which is to a strong degree shaped by the epistemological assumptions grounding the actors. Papadopoulos goes so far as to suggest that the less examined the epistemological assumptions grounding the therapeutic positioning of the actors are, the more likely those assumptions will be rigid and rigidly pronounced with an encounter or response (2021).

Perhaps this provides a partial explanation for a current trend in creating 'integrated initiatives' between religious leaders and MHPSS professionals, as we have seen it, in which differences in therapeutic foundations between the two are mentioned briefly as a potential difficulty—the nature of those differences and difficulties remaining imprecise—and then the balance of time and focus is given to describing the best ways to introduce and inculcate religious actors into a selected psychosocial framework. The almost astonishing rigidness of this trend, in which one of two groups or parties being invited to collaborate are expected to offer themselves for absorption into the infrastructure of the other as a matter of course, certainly suggests that something has gone wrong at a foundational level. The integration of disparate viewpoints and the collaboration of diverse actors is a beautiful and ideal goal, but simple questions must always be asked in the very initial stages of planning towards this goal, i.e., integration on whose terms, using whose definition, and using which methodology?

### 3. A Trend in Context

In general, within the psychosocial literature, something similar may be seen. Although many studies, case reports, and position papers purport to combine or integrate religious and psychosocial themes to support efforts in vulnerable communities, they typically move into one of three positions. The first position seeks to prepare MHPSS professionals to adopt or gain competency with religious themes to provide better care for religious clients or populations (cf. Trusty et al. 2022; Pearson 2017; Abu-Raiya 2015). The second position seeks to soften religious actors—and especially indigenous religious leaders—to psychosocial concepts in the hope that they will foster higher levels of mental health within their communities (cf. Mbote et al. 2021; Anshel and Smith 2014; Levitt and Ware 2006, etc.). The third position seeks to create opportunistic openings in existing psychosocial paradigms to allow religious leaders and/or religious clients to treat religious themes and engage in religious sidebar during broader evidence-based interventions (cf. Knabb 2016; Sanderfer and Johnson 2016; Neff and MacMaster 2005, etc.).

Although all three positions seem to produce some good results, could any of the three be seen as an ideal integration of the therapeutic efforts of religious leaders and MHPSS professionals? From one perspective, all three could be seen to effectively marginalize religious leaders within the semblance of integration, to the degree that it is taken for granted that psychosocial concepts and assumptions must guide the overall efforts. In all three, the unique therapeutic potential that religious themes and religious actors offer is treated as supplemental to the potential of the psychosocial efforts being considered. The perspectives of religious leaders rarely pose a successful challenge in the literature to the psychosocial perspectives which dominate the 'integrated' efforts being undertaken. Additionally, the initiatives developing from all three positions are typically governed by MHPSS professionals, working in accordance with the epistemological frames which undergird the psychological discourse—frames which may be considerably distant from those employed by any religious actors invited to participate. Perhaps most telling, this situation is often microcosmically demonstrated in the literature by the 'integrated' contributions of religious leaders who themselves are also trained MHPSS professionals: the latter seems to provide the dominant structure of their arguments, while the former offers supplemental insights.

Should agencies and organizations interested in integrating the efforts of religious leaders and MHPSS professionals be content to create approaches which are dominated by psychological presumptions about human beings, health, wellness, suffering, and flourishing? At least on the surface, this may allow them to avoid the difficulties associated with bringing two different kinds of helpers together to collaborate on their own terms. Should they be content to offer local religious leaders elementary training to help them understand a few psychosocial concepts, so they can be assigned a role in larger, pre-existing MHPSS interventions designed for the community? MHPSS professionals like Bojana and religious leaders such as Andreanos, who work in proximity among vulnerable people and ostensibly wish to work together, are looking for something far richer. They are to some degree stuck in the recognition that they can neither abandon their own epistemological foundations nor pressure the other helper to abandon theirs—because each recognizes something in the other that they are missing.

### 4. What Is Missing?

There seems to be some consensus among MHPSS professionals and religious leaders as to the growing edges for religious leaders hoping to participate in multi-disciplinary helping enterprises; in fact, they can likely be summarized in three broad categories (Letovaltseva et al. 2023; Alexander 2023; Alexander and Deuster 2021; Alexander et al. 2020; Alexander 2020c). First, religious actors often need help orienting to each of the specific lines of effort involved in the wider enterprise, so that they understand who is present in the community, what shared language is used, what kinds of interventions are being used by the various helpers, and how all the lines of effort are designed to work

together. Without such an orientation, they can often feel unsure of how to relate to the outside helpers, skeptical of their activities, and avoidant towards invitations to dialogue and participation. Secondly, if religious leaders might be asked to work with religious survivors whose religious formation is different from their own, they may need help in understanding those differences better and perhaps some overall guidance on sensitive pluralistic engagement. Third, religious leaders are often interested in being offered one or more helping frameworks which can help them better structure their efforts, but which does not oblige them to learn an entirely new trade or compromise their core beliefs.

These three gaps are well-known and well-expressed in the literature and in fact are carefully addressed by key lines of effort in popular religious training paradigms for religious leaders preparing to care for religious person in crisis (cf. Alexander 2023; Lasair 2020; Wallace 2017; Lawrence 2017; Nieuwsma et al. 2017; Alexander 2016; Nieuwsma et al. 2014; Seddon et al. 2011; Mendenhall 2009; Holifield 2005; Jernigan 2000). In many cases, although these training paradigms are typically led by religious leaders, psychosocial professionals are called upon to participate in such paradigms, to offer knowledge and experience to religious participants. This is not surprising, given the developing theme of this paper. For the sake of mutuality, a parallel question might form: in what ways are religious leaders able to offer knowledge and experience to MHPSS professionals in the areas in which MHPSS professionals are most vulnerable, by the limitation of their training and contexts?

It is important to recall the quite serious warnings within the literature related to the limitations and dangers of the prevalent approaches to MHPSS interventions after community crisis. To begin with, these approaches have shown the stubborn tendency to pathologize the suffering of certain survivors whose pain and disorientation, while significant, does not fit into psychiatric triage or assessment categories (Papadopoulos 2020; Alexander 2020a; Alexander 2019; Baillot et al. 2013; Pupavac 2002). This phenomenon can be considered from a variety of perspectives. First, it seems most likely to occur when (a) response structures are under pressure due to large pools of applicants or patrons, (b) planners and managers express the need—or share donor or wider agency expectations, etc.—to achieve results quickly, forming an additional layer of pressure, and (c) the level and scope of services provided become directly dependent on screening for pathological conditions (cf. Alexander 2020b; Papadopoulos 2002; Clough 1997). Indeed, these conditions often coexist and powerfully align during and after acute crises. Under such conditions, MHPSS professionals are likely to feel that they have little time or space to consider individual reactions to adversity outside of whether and how these reactions fit into pathological categories which indicate predetermined levels and methods of intervention as dictated within their care structures.

In one sense, this is inevitable. After all, how can adequate MHPSS responses to wide-scale calamities, for instance, be developed without firm structures that privilege careful triage and assessment practices? Additionally following, if responses flow through careful triage and assessment practices, is it not predictable that care workers will find their general orientation to survivor distress shaped by the assessment categories, as they separate survivors into two broad groups of those who 'need assistance' and those who do not? However, the apparent inevitability of these conditions in a care response does not relieve agencies or societies from anticipating likely second order effects which may have negative impact on a survivor population.

For instance, it is well documented that systemic pressures are created under such conditions which undergird movement within survivor populations towards the medical pathologization of their distress—i.e., when they have a vested interested in receiving higher levels of benefit and may be happy to have their experiences classified in whatever category will drive that higher benefit (Papadopoulos 2021; Freedman 2010; Papadopoulos 2000; Swartz and Levett 1989). Relatedly, the observation that care workers are likely to employ the lens of assessment and its categories beyond initial assessment is also widely documented (Papadopoulos 2021; Freedman 2010). What, then, is the impact of such effects

on the total MHPSS response, when survivor distress is most often assessed by searching for symptoms categorically associated with psychiatric disorder, and yet the number of survivors of any type of adversity who develop a psychiatric disorder are dwarfed by the number of survivors who—while they may suffer considerably—do not?

Enter the trauma discourse. The dominance of the application of the term 'trauma' to a nearly unnumerable variety of circumstances, events, and responses related to human adversity is posing a threat to survivors that is becoming increasingly clear across the field. Although a fuller discussion of this multi-faceted situation is beyond the scope of this article, the following three threats may be seen as especially salient to the concerns we have already addressed: (1) the apparent widespread conflation of traumatic reactions with associated events, birthing the new concept of 'traumatic events', which encourages even casual observers to begin to make the preposterous assumption that a particular type of event will inevitably lead to the traumatization of all survivors, (2) the widespread confusion of the term 'trauma'—which is now used to describe a startlingly broad spectrum of phenomena—with the psychiatric category of Post-Traumatic Stress Disorder (PTSD), and (3) the widespread report of large numbers of PTSD among survivors of war and disaster which are collected solely by surveys and inventories that in many cases rely upon self-assessment, even though PTSD is a complex psychiatric diagnosis which in many countries can only be formally diagnosed by an experienced psychiatrist or clinical psychologist.

In our experience, the way MHPSS professionals deployed to crisis areas are often trained and predisposed to employ and rely upon the term 'trauma'—i.e., not only as a part of their initial assessment orientation to survivors, but in a broad and enduring categorization of two groups of survivors: those who are 'traumatized' who need very particular interventions and those who are 'non-traumatized' or 'resilient' who likely need none—is cause for considerable concern. This situation tends to remove survivors whose suffering is not consonant with the assessment criteria from the vision of caregivers and to lock caregivers and the rest of the survivors into a frozen position which is focused primarily on damage—and damage of a certain kind—and which neglects longer arcs of experience within which suffering can be seen to contribute to long-term growth.

So, along with the ever-existing dangers related to pathologizing and oversimplifying the experiences of survivors, there emerges this danger related to preserving a balanced view of human suffering. In nearly every culture on earth, a phrase of this kind can be found: 'what does not kill you strengthens you.' Indeed, the authors themselves and likely everyone reading this article can attest personally to this maxim and recall that some of the most important catalysts for growth in their lives were activated through suffering. *Telos* is a Greek word that refers to an ultimate end, but this end is essentially bi-dimensional (Papadopoulos 2006). Every time of illness and crisis has the capacity to degrade and even destroy many positive qualities of the persons involved. Inevitably, every illness and crisis also bring the potential to enhance certain positive qualities of the people involved, even creating the conditions for the emergence of new positive qualities which were not previously present. In fact, the Greek word *crisis* contains these simultaneous potentials, and this is quite usefully preserved in the medical definition of crisis as a crucial or decisive point in an unstable situation, tending either towards improvement or deterioration (Merriam-Webster 2016).

Again, most MHPSS responses to crisis are pathologically focused, and for good reason. This fact mirrors the internal focus of most survivors of adversity, who are almost uniformly preoccupied with their pain, the erosion of their health and livelihood, and the potential further loss of both. However, it is the ideal role of caregivers to aid survivors and communities to achieve a more balanced view, to ensure human flourishing from a wider perspective. During crisis, people do suffer the loss of some of their positive qualities, and this is not to be minimized. Additionally, however, during times of crisis, people can sometimes become more generous or grateful than they were before. People can come awake to their realities in a way they were not before. They often refresh their priorities,

make new resolutions, reach out to estranged friends and family members, and realign with their highest sense of meaning and their ultimate values. Some ask the big questions of life, and the pursuit of such questions may have consequences that last until the end of a person's life, perhaps decades later.

A teleological approach to suffering, therefore, is one which does not reduce suffering to its imminent, physiological, or even intrapsychic pain dimensions. Rather, a teleological approach to suffering—while not ignoring or minimizing pain—also considers the potential growth that may emerge from this pain or may already be emerging in the pain. The fact that this wider approach to suffering is far more likely to be preserved within storied communities than within therapist–patient dyads is well demonstrated in the literature, even when the therapist is systemically trained (e.g., Sedikides and Wildschut 2018; Portova 2013; Kostic et al. 2005; Hollander 1999). We have, in fact, argued elsewhere that offering interventions from a teleological approach to suffering may be seen as a primary hallmark of spiritual care across many world religious traditions and that such interventions may provide an important complement or even an important corrective to dominant MHPSS paradigms (Alexander 2023; Letovaltseva et al. 2023).

This provides a transition to one final consideration, in examining shortcomings in prevalent MHPSS approaches to community crisis: under crisis conditions, there are many intangible aspects of survivor experiences which are not adequately addressed within psychosocial paradigms, precisely because of their intangibility. Here, we are using the term 'intangible' in a technical sense, referring to the near imperceptibility of such aspects and their impact on survivor identity (cf. Rasic 2021; Denes 2015; Bodin and Tengo 2012). One of the great benefits of the psychosocial discourse is that it seeks to grasp the totality of the person, i.e., the 'whole' person in the 'whole' of a person's environment. However, the totality of a person is enormously complex and at times quite elusive! For this reason, some pioneers within the discourse have occasionally attempted to add new prefixes or suffixes to the already compounded term to account for more of the totality, i.e., bio-psychosocial, bio-psychosocial-spiritual, etc. (cf. Neff and MacMaster 2005). The danger certainly does not lie in the fact that many MHPSS discourses fail to *aim* to capture the totality of the person. This is not only commendable but far superior to the alternative. The danger rather lies in the impossibility of the full success of the aim when this impossibility is not recalled constantly.

For instance, MHPSS assessment typically attempts to capture distress related to separation from loved ones and major changes in social roles. However, dislocated persons also can suffer from more intangible disruptions, i.e., in the familiar sights, sounds, smells, rhythms, movements, sensations, architecture, cycles of time, and rituals which—in constant interaction with the dynamics of their relationships and various identities—have long formed a complex backdrop to their lives. This backdrop serves as a key component of a larger sense of stability and predictability which is essentially taken for granted until it is compromised, resulting in a fundamental sense of unsettledness that can both cause significant disorientation and elude clinical assessment (Papadopoulos 2022; Papadopoulos 2021; Alexander 2020a). This is one of many such examples, and the existence of such examples do not by themselves ultimately undermine the psychosocial discourse. Being reminded of such examples, however, is important to everyone in the field; this to never lose sight of its limitations. Indigenous religious leaders are often in an ideal position to notice and to interact with such intangible themes in the care that they offer—once again offering the potential to offer an important complement or corrective to prevalent MHPSS paradigms.

At the beginning of this section, three growing edges were listed for religious leaders interested in participating in broader multi-disciplinary helping efforts, along with the claim that these growing edges are well-represented together in the literature. Now, a summary may be made of three significant growing edges common to MHPSS responses after humanitarian crisis, which are also clear—though perhaps not often stated together—in the literature. Although prevalent MHPSS paradigms admirably pursues the ideal of



caring for the whole person, in the whole of the person's environment, (1) the pressures converging on MHPSS professionals in crisis response often create the conditions for their assessment categories to radically narrow and oversimplify their vision, limiting their ability to remain open to the complexity, uniqueness, and totality of the survivors they encounter; (2) these same pressures create the conditions for the negative aspects of survivor reactions—and the negative potential of their suffering in general—to dominate the perspectives of MHPSS professionals, and (3) although many approaches to MHPSS training are broad, and attempts to draw in all of the perceptible aspects of the person for consideration, some powerful aspects of survivor experiences, are so intangible as to be nearly imperceptible and can therefore go completely unnoticed—or else can be forced awkwardly into more tangible but inadequate categories which frustrate understanding and helping efforts.

We the authors contend that religious leaders are often poised to aid MHPSS professionals in the exact areas of these three limitations above and that this activates a potential in crisis response for religious leaders and MHPSS professionals to truly work together in the spirit of mutuality, where the assumptions of each type of helper offers a positive and needed influence on the other. To demonstrate this potential, we offer a vignette.

### 5. Dinah's Story

A religious leader we will call Dinah was recently invited to serve a role on a humanitarian medical service dedicated to treating new asylum-seekers at an international border—the majority of whom share Dinah's religious foundations.[6] Before she deployed, Dinah was offered a training which purported to prepare religious leaders for service in MHPSS support roles. In fact, this training was built upon a central module during which a psychologist offered an introduction to 'trauma theory' and a simple framework for psychological first aid. This training left Dinah initially uncertain of her ability to contribute, because she was unfamiliar with most of the concepts in the training. When she shared this uncertainty in passing with a pharmacist who was a part of the team, the pharmacist warmly encouraged her to feel free to offer something from her own experience when in doubt. This reassured Dinah, and, while keeping the training materials for review, she decided to also trust her own foundations.

Soon after arriving to the border, Dinah was asked by MHPSS professionals to speak with a man named Hamna in that asylum-seeking community, who had been labeled as (a) influential to the community and (b) resistant to the team's assessment and care strategies. In other words, Hamna was functioning as a 'problem', a human barrier to the psychosocial team's efforts. In our experience working with religious leaders responding to crisis, Dinah's first intervention was one that many religious leaders would employ in a situation where someone has been labeled as a 'barrier', or a 'problem'. The intervention began within herself, as an attempt to restore complexity to her own vision of this man as a unique human person. She wondered: perhaps in some way Hamna is a 'problem' when viewed solely through the lens of a planned intervention, but what else is he? What other lenses can our larger team employ, to see him in the fullness of his humanity in this unique context? Certainly, she thought, a human being cannot be reduced to a 'problem'. In this act of epistemological vigilance, she was able to help the team to reconsider their frustration and their considered courses of approach to Hamna.

In a conversation with the MHPSS team leader, Dinah mentioned a wider discomfort she felt was arising from the triage process. She felt that the team—having become accustomed in the first days of deployment to performing quick, sweeping decisions about how to place people into various initial care categories—had begun to make a new set of sweeping decisions about which survivors were 'resistant' and 'compliant'.

To his credit, the MHPSS team leader took Dinah's concerns seriously and allowed her to speak briefly to both incoming and outgoing workers at an upcoming shift change. Dinah used subtle humor, reminding everyone present that of course, no human in any situation is 100% resistant or 100% compliant or 100% 'anything'. She said that as a young

girl she had learned in her spiritual reading that when people are reduced to something less than human, something much larger has developed between people, often with roots in fear or misunderstanding. As the deployment developed over two weeks, the MHPSS team leader realized that, as a religious contributing to the larger effort, Dinah also had the luxury of time and space to think at various levels that her colleagues were not always able to manage, and he asked for her perspective on a regular basis.

Dinah also spent time with patients that the MHPSS team found difficult—and this had the dual effect of winning her the respect and gratitude of the team and the opportunity to engage in some of her richest encounters during the deployment. Dinah began meeting with a small group of men and women, functioning as a type of extended family, which included Hamna. She was stunned to discover that, in his home region and country, this man was a former poet laureate. He was a cultural leader among his people. Although he was still with his people, he felt he had left his position of prominence among them when he joined them in powerlessness, to flee their common homeland. He felt relief to some degree, in receiving care at the border. Simultaneously, he experienced a deep skepticism about receiving health and MHPSS interventions he had never encountered before. He also experienced a resentment at not being able to help and lead his people the way he wished. He also experienced a drive to protect the people around him, lest they be mistreated in a vulnerable situation. In summary, his resistance to the interventions was quite complex, and from Dinah's perspective, it represented a previously hidden opportunity. She even became quite convinced that some of the reason that Hamna frustrated caregivers was that he spoke in poetic language.

In our experience, is quite common for MHPSS professionals to either dismiss poetic language as technically irrelevant or else to attempt to 'migrate' such language into the terminology of professional frames. However, poetic language often contains clues that—if taken within their widest ecological context—can be quite relevant, even if they serve to turn the attention of caregiver outside of their technical preoccupations. When encountering poetic language in crisis, religious leaders are far less likely than other helping professionals to miss this language, to avoid it, or to attempt to migrate it into a different discourse. Dinah brought her discovery to the MHPSS team leader, along with a recommendation that some possibility be given to allow Hamna and perhaps others in the community to begin meeting to recite their poetry, and to engage in other creative expressions such as sharing of sacred stories, community history in the oral tradition, and traditional song and dance. She suggested that, in their shared religious tradition, this is a crucial part of how meaning is made and maintained, how suffering is placed into longer and broader context, and how identities are stabilized in hard times.

Dinah was not able to extend her deployment to witness how and whether these suggestions were implemented, but it is doubtless that the potential of her suggestions and the importance of her presence on the team to make such suggestions is difficult to overstate. It can clearly be seen in this vignette how a religious actor, operating from a unique set of core assumptions and a unique approach to care, might serve both as a protective agent and a catalyst for growth among MHPSS professionals in the aspects of care which represent the greatest growing edges of prevalent MHPSS paradigms at large.

## 6. A Vision for Integration, or an Integrated Vision?

It is our contention that the wisest and most successful attempts to integrate the efforts of religious leaders and MHPSS professionals responding to humanitarian crises will begin with close resemblance to the example offered by Bojana and Andreanos. Rather than avoiding each other or forcing Andreanos to essentially work under the assumptions and conditions of Bojana's orientation, they explored their core differences and considered how these core differences might represent opportunities to bolster the growing edges they each had, corresponding to limitations of their own distinct positions and approaches in care. This example of mutuality is not common, in our experience, but it could be. Even in trainings offered by MHPSS professionals to religious actors, designed to help religious

actors in their growing edges, much clearer guidance could be provided on the limitations of prevalent MHPSS paradigms and the potential for religious leaders to provide support which counterbalances these limitations. As much as it is possible to employ training teams which feature both religious leaders and MHPSS professionals who are experienced in offering crisis care from their own orientations, the better the chances will be for any truly mutual collaborations to develop.

If such an auspicious beginning in training is not possible, it is our further contention that true collaboration might occur anyway, as demonstrated in the development of Dinah's role in helping the MHPSS team at an international border. It is doubtless that Dinah received some instruction before deployment that helped her, i.e., oriented her to the work of the MHPSS team and its place in the wider helping enterprise in that particular setting and equipped her with an elementary psychological first aid model. However, it is also doubtless that Dinah oriented her MHPSS colleagues to certain dangers arising in their work, related to an oversimplified vision of survivors shaped by assessment categories, a tendency to overfocus on negative aspects of adversity and a difficulty appreciating less tangible aspects of survivor experiences. In this she served as a protective agent in the overall effort, protecting her team and, most importantly, protecting survivors also.

**Author Contributions:** Writing—original draft, D.W.A. and T.L. All authors have read and agreed to the published version of the manuscript.

**Funding:** This research received no external funding.

**Institutional Review Board Statement:** Not applicable.

**Informed Consent Statement:** Not applicable.

**Data Availability Statement:** Not applicable.

**Conflicts of Interest:** The authors declare no conflict of interest.

## Notes

[1]   Bojana's name was changed to protect her identity. This vignette is crafted from a personal communication recorded by work journal entry in September 2016 (unpublished).

[2]   In this paper the authors appeal to a standard legal-political definition of 'traditional religious leader', i.e., an individual member of a group or community who is recognized by other members as an expert in the group's traditional religious beliefs and practices (cf. South Africa's Municipal Structures Act 117 of 1998, as an example).

[3]   Andreanos' name was changed to protect his identity. This vignette is crafted from a personal communication recorded by work journal entry in August 2016 (unpublished).

[4]   This information is derived from personal communication between Bojana, Andreanos, and the authors (Alexander), recorded in a work journal in August-September 2016, and again in September of 2017 and September of 2018. Because they could not both agree to have their names and the details of their work recorded here, both agreed to have certain details of their relationship shared in this paper, without direct attribution.

[5]   Personal communication between Bojana, Andreanos, and the authors (Alexander), recorded in a work journal in August-September 2016, and again in September of 2017 and September of 2018.

[6]   Dinah is a unique person, and her story was related to one of the authors (Alexander) in 2019 and recorded in his work journal. However, certain aspects of her situation are changed in this article, for the primary purpose of ensuring her anonymity.

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
