# Peer review of "Psychosocial Workers and Indigenous Religious Leaders: An Integrated Vision for Collaboration in Humanitarian Crisis Response"

_religions, doi:10.3390/rel14060802_

Round 1

Reviewer 1 Report

very interesting and practical article for psychosocial worker

Author Response

Thanks for your time and help in this endeavor.

Reviewer 2 Report

The strength of the article is within the vignettes which are well written. The intention of the paper is convincing. However, while the first part of the paper is very strong in describing current research and understanding on the importance to integrate religious leaders' approach and MHPSS professionals' approach, the second part is less precise and documented by literature. The attribution of a teleological understanding of suffering to (indigenous) religious approaches is, in my view, not adequate or convincing. 

While the paper is strong in describing MHPSS strategies and approaches to survivors and situations of crisis, it lacks a description of (indigenous) religious leaders' care and interventions. Dinah's vignette is a great example of a chaplaincy approach not too much focused on ritual but on deep listening, hermeneutical understanding, and building trust-relationships. Are these specifics of religious and spritual care? Here, the aoutors could be more precise.  

Very small changes needed.

However: please make sure that all names an pseudonyms. The name "Vasilios" appears twice, and is seems to be the clear/real namen of the person involved. 

Author Response

Thank you very much for your time - I am making the adjustments you suggest.

Reviewer 3 Report

It is not clear what do the authors mean by "tensions" between psychology and religion? Do they refer to religion-science conflict (psychology being a branch of science here)?

I found the story of Bojana distracting. This kind of writing is not common in those fields that I am familiar with.  And the more I went through this paper the more I'm convinced that this is not an original research paper that deserves publishing in this journal, unless it has been submitted as a "letter to editor" or something like that.

Author Response

Thank you very much for your time.  Yes, I was referring to the tensions in the literature (between psychology and religion) as pronounced in seminal work over the past 7-10 years such as Hodge et al., 2020; Abu-Raiya, Pargament, & Krause, 2016; Paloutzian & Park, 201.  The tension is an essential / epistemological one, in my view.  I will attempt to make this clearer.